# Crosstalk between the Hippo Pathway and the Wnt Pathway in Huntington’s Disease and Other Neurodegenerative Disorders

**DOI:** 10.3390/cells11223631

**Published:** 2022-11-16

**Authors:** Pasquale Sileo, Clémence Simonin, Patricia Melnyk, Marie-Christine Chartier-Harlin, Philippe Cotelle

**Affiliations:** 1Univ. Lille, INSERM, CHU Lille, UMR-S 1172, Lille Neuroscience and Cognition Research Center, F-59000 Lille, France; 2Centre de Référence Maladie de Huntington, CHU Lille, F-59000 Lille, France; 3ENSCL-Centrale Lille, CS 90108, F-59652 Villeneuve d’Ascq, France

**Keywords:** Hippo, YAP, Wnt, Huntington’s disease, neurodegeneration

## Abstract

The Hippo pathway consists of a cascade of kinases that controls the phosphorylation of the co-activators YAP/TAZ. When unphosphorylated, YAP and TAZ translocate into the nucleus, where they mainly bind to the TEAD transcription factor family and activate genes related to cell proliferation and survival. In this way, the inhibition of the Hippo pathway promotes cell survival, proliferation, and stemness fate. Another pathway can modulate these processes, namely the Wnt/β-catenin pathway that is indeed involved in cellular functions such as proliferation and cell survival, as well as apoptosis, growth, and cell renewal. Wnt signaling can act in a canonical or noncanonical way, depending on whether β-catenin is involved in the process. In this review, we will focus only on the canonical Wnt pathway. It has emerged that YAP/TAZ are components of the β-catenin destruction complex and that there is a close relationship between the Hippo pathway and the canonical Wnt pathway. Furthermore, recent data have shown that both of these pathways may play a role in neurodegenerative diseases, such as Huntington’s disease, Alzheimer’s disease, or Amyotrophic Lateral Sclerosis. Thus, this review analyzes the Hippo pathway and the Wnt pathway, their crosstalk, and their involvement in Huntington’s disease, as well as in other neurodegenerative disorders. Altogether, these data suggest possible therapeutic approaches targeting key players of these pathways.

## 1. Introduction

Altered homeostasis is observed in many neurological disorders such as Alzheimer’s disease (AD), Parkinson’s disease (PD), Huntington’s disease (HD), and Amyotrophic Lateral Sclerosis (ALS). Despite differences in etiologies, symptoms, and sites of pathology between these neurological disorders, a number of shared pathways have been identified for many or all of these diseases including Hippo and Wingless-Int (Wnt) pathways. These two signaling pathways play crucial roles in maintaining tissue homeostasis by orchestrating cell proliferation, differentiation, and apoptosis.

The Hippo pathway consists of a cascade of kinases that finally controls the phosphorylation of Yes-associated protein (YAP) and a transcriptional coactivator with PDZ-binding motif (TAZ), hence their cellular localization. Unphosphorylated YAP and TAZ enter the nucleus and mainly interact with transcription factors known as TEA domain family members (TEAD) (in mammals, four TEAD genes encode for four homologous members, TEAD1–4), promoting cell survival, proliferation, and stemness. It follows that YAP/TAZ, and, in general, the Hippo pathway, play crucial roles in various types of cancer, as widely demonstrated so far and explained in detail in several reviews [1,2,3,4].

Wnt (Wg, Wingless and Int, Integration site) is a family of glycoproteins that activate three different pathways: the canonical Wnt/β-catenin cascade, the noncanonical planar cell polarity (PCP) pathway, and the Wnt/Ca^2+^ pathway [5,6,7]. In this review, the focus will be only on the canonical Wnt/β-catenin cascade, given the implications it has with HD and the Hippo pathway (see below); therefore, for an overview of the non-canonical pathways, please refer to reviews focused on these topics [8,9].

In HD, several symptomatic treatments have been used for decades, but many trials of potential modifier treatments focusing on neuron degeneration processes have failed [10,11]. More recently, huntingtin-lowering therapies acting more upstream in the disease pathophysiology are in development, but the first two trials (generation HD1 (NCT03761849) and precision-HD1 (NCT03225833)) using intrathecal ASO were stopped for safety reasons.

Many compounds that have been demonstrated to target the Wnt pathway are under investigation in clinical trials in AD, PD, and ALS [12], but not in HD, and to date, no molecules targeting the Hippo pathway have reached the preclinical stages. In this context, the purpose of this review is to discuss the connection of the Hippo pathway with the Wnt pathway in Huntington’s disease and other neurodegenerative disorders and pave the way to new potential therapeutic options.

## 2. Hippo Pathway

The most important known functions of the Hippo pathway are to promote cell survival and proliferation but also differentiation and migration in developing organs [1,13,14,15,16]. This pathway is regulated by different mechanisms, such as cell–cell contact, cell polarity, and actin cytoskeleton but also by different signals such as cellular energy status, mechanical cues, and hormonal signals that act through G-protein-coupled receptors (see below). The core of the Hippo pathway in mammals is a kinase cascade (Figure 1), mainly represented by two pairs of kinases: the mammalian Ste20-like kinases 1/2 (MST1/2), which phosphorylate and activate another kinase large tumor suppressor 1/2 (LATS1/2). The role of these kinases is to restrict or inhibit, by phosphorylation, the activities of the two transcriptional coactivators, YAP and TAZ. When YAP and TAZ are not phosphorylated, they translocate into the nucleus, where they mainly bind to the TEAD transcription factor family and activate genes related to cell proliferation, survival, and migration [1,16,17,18,19,20,21,22]. In addition to proteins of the TEAD family, YAP and TAZ may interact with other transcription factors such as RUNX1/2 and TBX5 [1]. However, when the Hippo pathway is active, the kinase cascade leads to the phosphorylation of YAP and TAZ, which are excluded to the nucleus, sequestrated into the cytoplasm, and/or degraded by the proteasome [23,24,25,26]. The phosphorylation and the consequent activation of MST1/2 can be performed by the TAO kinases (TAOK1/2/3). These kinases then phosphorylate the activation loop of MST1/2 (Thr183 for MST1 and Thr180 for MST2) [27,28]. Alternatively, the phosphorylation of the activation loop can be achieved by MST1/2 autophosphorylation [29], and then the activation loop phosphorylation is enhanced by MST1/2 dimerization [30]. When MST1/2 are phosphorylated, consequently they phosphorylate SAV1 and MOB1A/B [31,32]. The latter are two scaffold proteins able to cooperate with MST1/2 in the recruitment and phosphorylation of LATS1/2 at their hydrophobic motifs (T1079 for LATS1 and T1041 for LATS2) [33,34]. Another important factor involved in LATS1/2 phosphorylation is NF2/Merlin, which directly interacts with LATS1/2 and facilitates LATS1/2 phosphorylation by the MST1/2–SAV1 complex [34]. Once LATS1/2 is activated, it undergoes autophosphorylation and inactivates YAP and TAZ [26]. The phosphorylation of YAP and TAZ leads to their binding with 14-3-3, and this binding causes the cytoplasmic sequestration of YAP/TAZ [26]. Furthermore, LATS-induced phosphorylation could induce the phosphorylation of YAP/TAZ by Casein kinase 1δ/ε and the recruitment of ubiquitin ligase SCF E3, consequently leading to the ubiquitination and degradation of the proteins [24,25]. Interestingly, YAP can also be degraded by autophagy [35]. YAP and TAZ do not have DNA-binding domains; therefore, in the nucleus, they interact with TEAD1–4, which are sequence-specific transcription factors. In this way, they regulate the transcription of the target genes [19].

Besides YAP/TAZ, the most-studied coactivators of TEAD1–4 transcriptional activity are the Vestigial-like (VGLL) protein family, which consists of four members (VGLL1-4). These members interact with TEAD to regulate gene expression. VGLL1-4 bind to the YAP–TAZ binding domain of TEAD1-4, VGLL1-3 and VGLL4 have distinct molecular functions, and only VGLL4 has been reported to inhibit YAP/TAZ–TEAD interactions [36]. The interaction with YAP/TAZ allows TEAD1–4 to dissociate from VGLL4 and then activates TEAD-mediated gene transcription to promote tissue growth and inhibit apoptosis [37]. YAP also competes with VGLL3, and inhibiting its phosphorylation constitutes a new therapeutic option for the treatment of endocrine-resistant breast cancers [38].

There are several upstream signals that regulate the Hippo pathway such as cell–cell contact, mechanical cues, and hormonal signals. For example, β-catenin is a linker between cadherins and the actin cytoskeleton that inhibits YAP activity in keratinocytes. The disturbance of the E-cadherin/β-catenin complex will then decrease YAP phosphorylation and consequently increase nuclear YAP localization [39,40]. The most important mechanisms that regulate YAP/TAZ activity are mechanical and cytoskeletal signals. When the cell grows and can stretch on the extracellular matrix (ECM), the cytoskeletal adaptation to the cell spread shape leads to the activation of YAP/TAZ and their nuclear accumulation, promoting cell proliferation and inhibiting differentiation [41]. On the contrary, when the cell grows on a small ECM adhesive area, the shape is round and compact. In this context, YAP and TAZ do not enter into the nucleus and cells stop to proliferate and initiate differentiation. In addition, changes in the ECM stiffness affect YAP/TAZ localization. When the ECM is rigid, YAP and TAZ are active in the nucleus, while when ECM is soft, YAP/TAZ are inactivated [41]. Moreover, the tissue architecture could act on YAP/TAZ regulation inhibiting cell growth and proliferation. Indeed, during cell growth at low density, YAP and TAZ are nuclear and active. However, at a high cellular density, YAP/TAZ become cytoplasmic [26]. YAP and TAZ phosphorylation are increased during the contact inhibition of proliferation, suggesting thus activation of the Hippo pathway. Cell–cell contact at a high cell density produces a growth inhibitory signal mediated by the Hippo pathway [17,26,42]. This is due to the fact that when the cell-contact increases, the E-cadherin/catenin system triggers LATS1/2 activation and YAP/TAZ phosphorylation. This phenomenon contributes to about 30% of growth inhibition by acting on YAP/TAZ relocalization and leaving cells with the right amount of YAP/TAZ in the nucleus to continue proliferation [1,43]. In other words, when adherens junctions and tight junctions increase in confluent cells, they contribute to the activation of LATS1/2 and the phosphorylation of YAP and TAZ [26,44].

Regarding the regulation by circulating molecules, numerous studies have shown that the regulation of the Hippo pathway by GPCRs is a response of cells to hormonal cues and many GPCRs signal through YAP/TAZ [45,46,47,48,49]. It has been shown, for instance, that Gαs-coupled signals (i.e., glucagon, epinephrine) repress YAP/TAZ, while lysophosphatidic acid (LPA) and sphingosine-1-phosphate (S1P) activate and stabilize YAP and TAZ through their G protein-coupled receptors (GPCRs), LPA receptor (LPAR), and S1P receptor (S1PR) [50]. Therefore, GPCRs, based on stimuli and ligands, can activate or inhibit YAP. Mechanistically, Rho-GTPases, being part of the transduction cascades activated by GPCRs, regulate YAP/TAZ phosphorylation [51,52]. Precisely, Gα12/13- and Gαq/11-coupled GPCRs activate Rho-GTPases, which inactivate LATS1/2 [50]. Surprisingly, the metabolic pathway, initiated by the mevalonate/HMG-CoA reductase, also supports YAP/TAZ activity, increasing cell proliferation and migration [53,54]. As mentioned above, even stress signals can regulate the Hippo pathway. It has indeed been shown that MST1/2 are activated by hydrogen peroxide, and they have a role in cellular oxidative stress responses [55,56]. Furthermore, YAP binds to FOXO1 activating a catalase reaction and genes in order to reduce oxidative stress [57], suggesting a function of YAP in reactive oxygen species (ROS) scavenging. eIF2αP plays a pivotal role in the regulation of redox homeostasis and the adaptation of eukaryotic cells to oxidative stress. eIF2αP activates the Hippo pathway to promote death in response to oxidative stress through the stabilization of LATS1 by ATF4 [58]. Other stress conditions act on the Hippo pathway such as glucose deprivation, which induces YAP and TAZ phosphorylation by LATS1/2 activation [59]. Finally, under energy stress conditions, AMP-activated protein kinase (AMPK) can directly phosphorylate YAP, interfering with the interaction of YAP–TEAD and inhibiting the transcription of the related genes activated by them [60,61].

## 3. Wnt/β-Catenin Pathway

The Wnt/β-catenin signaling pathway is a cell survival pathway that is mostly involved in animal development, cell fate, cell proliferation, differentiation, and tissue regeneration and also has a central function in disease processes and tumorigenesis [62,63,64,65,66,67,68,69]. In humans, there are 19 Wnt proteins [70], which are translated in the endoplasmic reticulum (ER), where they are O-palmitoylated by an enzyme named porcupine (palmitoleate at Ser209 (Wnt3a)) [71]. This post-translational modification is an essential step for the secretion and activity of Wnt proteins [72,73,74,75] and thus their secretions. Palmitoleate-CoA is generated by the action of stearoyl-CoA-desaturases (SCD-1) [76]. In the extracellular medium, O-palmitoleated-Wnt can be inactivated by Notum, an enzyme able to deacylate Wnt [77]. The Wnt-β-catenin signaling pathway constantly regulates the synthesis and degradation of β-catenin by post-translational modifications, controlling in this way the levels of cytosolic β-catenin (Figure 2). In the absence of Wnt ligands, the pathway is not active, and this leads to keeping the cytoplasmic levels of β-catenin low via ubiquitin-dependent proteasomal degradation [78]. This degradation is regulated by the β-catenin destruction complex [79], which is composed of the scaffolding proteins Axin [80,81] and adenomatosis polyposis coli (APC) [82,83], and the kinases CK1α and GSK3α/β [84,85,86]. The destruction complex proceeds by phosphorylating the β-catenin [87], thus allowing its recruitment to the SCFβ-transducin repeat-containing protein (β-TrCP) E3-ubiquitin ligase and the consequent proteasome-mediated degradation [88,89,90]. Instead, when Wnt ligands are present, they bind to the receptor, Frizzled (FZD, a seven-transmembrane receptor) [91,92,93], which forms a heterodimer with one of the Wnt co-receptors, LRP5 or LRP6 (LRP5/6) [94,95,96]. Once bound by Wnt, the FZD-LRP5/6 complex activates the canonical signaling pathway and recruits the cytoplasmic protein, Dishevelled, interacting directly with Frizzled and with Axin-GSK3 complexes via Axin–Dishevelled interaction [97,98,99,100]. Then, GSK3 and the Cdk14(PFTK1)-Cyclin Y mitotic kinase complex phosphorylate the intracellular region of LRP5/6 [101,102], precisely, the PPPSP motifs, allowing further phosphorylation by the membrane-anchored kinase, casein kinase 1γ [103,104]. Once phosphorylated, LRP5/6 blocks β-catenin degradation through the direct inhibition of GSK3 activity and sequestering it [105,106,107,108]. In this way, β-catenin accumulates in the cytoplasm. It can then translocate into the nucleus, replace repressor protein Groucho from transcription activator LEF/TCF [109,110], and activate Wnt-responsive genes in a lineage/type-dependent manner [111,112,113,114].

Azzolin et al. demonstrated that YAP and TAZ are also components of the β-catenin destruction complex [115]. YAP/TAZ can be sequestered in the cytoplasm where they interact with the destruction complex and associate with Axin. These proteins are involved in the recruitment of β-TrCP to the complex. YAP/TAZ seem to have an important role in β-catenin degradation when the Wnt is not active. Indeed, when YAP/TAZ are depleted, the β-catenin/TCF transcriptional response is active. Indeed, when the Wnt ligands are present, they lead to the association between the Wnt receptors LRP6 and Axin and the consequent release of YAP/TAZ from the destruction complex [115]. In this way the destruction complex cannot recruit β-TrCP, leading to β-catenin accumulation, and in the meantime, YAP/TAZ are free to accumulate in the nucleus and activate their dependent transcriptional responses. The YAP/TAZ cytoplasmic sequestration is therefore also assisted by their incorporation into the destruction complex in the Wnt-OFF cells. Interestingly, YAP/TAZ, in addition to acting as Wnt/β-catenin signaling antagonists in the cytoplasm, can also act as nuclear transcriptional mediators of Wnt signaling. The presence of YAP/TAZ in the cytoplasm can prevent β-catenin from entering the nucleus and counteract the phosphorylation of the Wnt transducer Disheveled [116,117]. Moreover, the destruction complex assembles a phospho-β-catenin/TAZ/β-TrCP association that leads to TAZ (but not YAP) degradation [118]. In confluent untreated cells, YAP and TAZ are mostly in the cytoplasm, while after Wnt treatment, the nuclear localization increases. In parallel, the activation of TEAD and the expression on YAP/TAZ target genes such as Cyr61 increases. Wnt treatment leads to the dissociation of YAP/TAZ from the destruction complex and to the parallel increase in Axin1-LRP6 association, suggesting that YAP/TAZ-Axin binding is outcompeted by LRP6. Moreover, the depletion of YAP/TAZ greatly reduces the association of β-TrCP with the complex while it has no effect on the Axin1, GSK3, and β-catenin interaction, suggesting that YAP/TAZ are needed for β-TrCP association to the destruction complex, and the phosphorylation of β-catenin alone is not sufficient [115]. Another study found that TAZ is a negative regulator of Wnt/β-catenin signaling [117]. In fact, the knockdown of TAZ increases endogenous β-catenin protein levels, increases the nuclear accumulation of β-catenin after Wnt3A stimulation, and enhances Wnt3A-induced Axin2 expression. Moreover, TAZ interacts with all three isoforms of Dishevelled (DVL), especially with DVL2. In particular, TAZ inhibits the Wnt3A-induced phosphorylation of DVL2. Furthermore, the phosphorylation of TAZ by MST/LATS led to the retention of TAZ in the cytoplasm, the inhibition of DVL action leading to the inhibition of Wnt/β-catenin signaling [117]. In addition, the Hippo pathway has been found to negatively regulate Wnt signaling. Imajo et al. found that the overexpression of the YAP/TAZ β-catenin induced the activation of T cell factor (TCF) transcriptional activity without suppressing the stability of β-catenin and without decreasing its protein levels [116]. These data disagree with the previous data of Varelas, in which TAZ suppressed the stabilization of β-catenin. Imajo et al. showed also that YAP and TAZ are able to bind to β-catenin, and the activation of the Hippo pathway suppresses nuclear β-catenin signaling. Moreover, the presence of β-catenin in the nucleus is reduced in cells exhibiting cytoplasmic YAP, and YAP suppresses Wnt/β-catenin signaling, not only in culture cells but also in Xenopus embryos. In another study, Azzolin et al. showed that Wnt stabilizes TAZ/Wnt pathway activation by stimulation with Wnt3a or by the intracellular inhibition of the destruction complex, which stabilizes TAZ and induces the expression of TAZ-dependent genes in a variety of cell types [118]. Moreover, the protein level of TAZ and the expression of its transcriptional response are increased with the depletion of β-TrCP. Additionally, the knockdown of β-catenin leads to the better stabilization of TAZ and an increase in TAZ transcriptional activity and nuclear accumulation. Interestingly, they showed that the “WW” domain of TAZ is responsible for binding with β-catenin, which acts as a bridge between TAZ and the β-TrCP complex [118]. Contrary to these studies, Heallen et al. showed that the presence of YAP in the nucleus together with β-catenin induces the expression of some genes of heart development [119]. Moreover, the knockdown of the Hippo pathway players induces the nuclear accumulation of β-catenin and the expression of its target genes and promotes the cardiomyocyte overgrowth phenotype in mice; this phenotype could be rescued by the heterozygous deletion of β-catenin. Interestingly this team also showed that β-catenin interacts in the nucleus with the nonphosphorylated form of YAP [119]. Therefore, an inhibitor of SCD1 was used to pharmacologically regulate the functions of YAP, TAZ, and β-catenin in lung cancer [120]. All these data indicate a close relationship between the Hippo pathway and the Wnt pathway; however, there are still many points to be clarified, some even contradictory, which require further study.

## 4. Hippo Pathway and Huntington’s Disease (Figure 3a)

Recent studies have shown the deregulation of several actors of the Hippo pathway in HD. Mueller et al. observed that neuronal nuclear YAP levels are decreased in human post-mortem brains and neural stem cells (NSCs) derived from HD patients. Moreover, they showed that there is no change in YAP transcript levels in HD, suggesting that the decrease is due to post-transcriptional events such as phosphorylation. In accordance with this, they found a significant increase in phosphorylated YAP (pYAP) levels in the human HD cortex compared to the control but no change in the total YAP levels. In parallel, they also observed an increase in pMST1/2, despite no change in the total MST1/2 level; this probably explains the increase in YAP phosphorylation and its consequent decrease in the nucleus in HD post-mortem brains. Due to the decreased nuclear YAP, they observed a decrease in the YAP/TEAD interaction. Furthermore, in the post-mortem human cortex, they found three other downregulated Hippo pathway genes (Lats2, Meis1, and Sav1) as well as a downregulation of the Hippo target gene Cyr61. Mueller et al. also obtained similar results in Htt CAG knock-in mice HdhQ111/Q111, where they observed a significant increase in pYAP and in pMST1/2 levels [121].

**Figure 3 cells-11-03631-f003:**
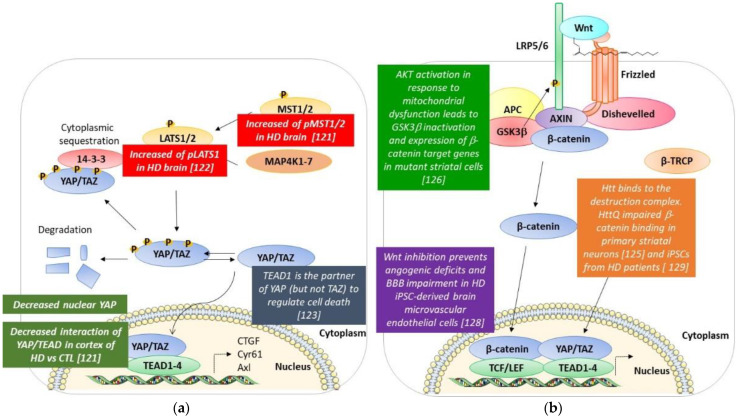
A schematic resume of the modifications of the Hippo and the Wnt pathways found in HD models and post-mortem HD brains. (**a**) Hippo pathway. (**b**) Wnt pathway. See text for details [121,122,123,125,126,128,129].

Similar results were obtained in another study by Yamanishi et al. They demonstrate that YAP is decreased in the nucleus of neurons in the cerebral cortex of human HD patients, and pLATS1 is increased in HD human neurons and mutant Htt-KI mice [122].

Yamanishi et al. have found a novel form of necrotic cell death called Transcriptional Repression-Induced Atypical cell Death (TRIAD) in HD human and mouse brains. TRIAD is characterized by multiple steps: (i) the destruction of the cistern structures of ER, (ii) the enlargement and vacuolization of ER, (iii) the detachment of the outer nuclear membrane from the inner membrane, (iii) the release of ER content into the cytosol, (iv) the ballooning of the cell body, the shrinkage of the nucleus, and enucleation [122,123]. They discovered that this form of necrosis is TEAD/YAP-dependent, and mutant Htt did significantly increase the incidence. As observed in the work of Mao et al., the knockdown of YAP promoted cell death, while the knockdown of LATS1/2 suppressed cell death. This indicated that nuclear YAP suppressed TRIAD, or that the deficiency of nuclear YAP promoted this atypical cell death. Further proof is that LPA and S1P, ligands of cognate GPCR receptors, which act to dephosphorylate YAP, suppressed cell death. On the contrary, stimulating GPCR receptors in order to activate the Hippo pathway and inactivate YAP led to the opposite effect. Moreover, it is important to note that the knockdown of TEAD1 accelerated cell death. Interestingly this team showed that the apoptosis promoter kinase named Plk1 phosphorylates YAP at Thr77, enhancing its interaction with p73 while decreasing the affinity to TEAD. Then, Plk1 is able to switch the partner of YAP from TEAD to p73, while LATS-mediated phosphorylation suppressed YAP interaction both with p73 and TEAD. In addition, when the authors analyzed the role of Htt in this regulation, they observed that Htt interacts with YAP with an increased affinity when the polyQ tract was expanded. Interestingly, it sequestrates YAP in the inclusion bodies. These data suggest that mutant Htt suppresses TEAD/YAP transcription through interaction with YAP as well as acting on the Hippo pathway, which ameliorates or suppresses TRIAD; the treatment of htt-KI mice with LPA and S1P resulted in the suppression of ER instability and cell death along with an improvement of the motor functions of mice [122,123].

All these results are very encouraging and have shown that acting on the Hippo pathway could represent an innovative therapeutic strategy to reduce HD phenotypes.

## 5. Wnt/β-Catenin and Huntington’s Disease (Figure 3b)

It is well established that the Wnt/β-catenin pathway is altered in HD models, especially the stability and levels of β-catenin. Studies have been performed in transfected cell models [124,125], striatal cells from murine models [126,127], the *Drosophila* model [125] iPSC-derived neuronal cultures [127,128,129], and finally, post-mortem striatal samples from HD patients [125].

Two studies show that the β-catenin level is decreased in HD. Firstly, Carmichael et al. observed a reduction in zβ-catenin level, associated with a reduction in TCF-mediated transcription in HD, and GSK-3 inhibition rescues polyglutamine-induced cell death in neuronal and non-neuronal cell lines (human neuroblastoma cells SKNSH and African green monkey kidney cells COS-7). The rescue is mediated by an increased level of β-catenin and its associated transcriptional pathway [124].

Ghotak et al. [127] found a significant decrease in the β-catenin level in cultured mutant huntingtin knock-in STHdh^Q111/Q111^ striatal cells. This observation led to decreased β-catenin/TCF-mediated transcriptional activity with the downregulation of the Wnt/β-catenin target genes. Interestingly, the downregulation of β-catenin is not governed by GSK3β-dependent proteasomal degradation. Conversely, these authors observed an increase in miR-214 more than ten-fold in HD compared to controls; this miRNA was previously documented to decrease the β-catenin protein level in hepatocellular carcinoma cell lines [130,131]. Interestingly, this downregulation of the β-catenin level is at the post-transcriptional level, and it is not mediated by GSK3β-dependent proteasomal degradation [127].

In the same cellular model, as well as in extracts of Hdh^Q111/Q111^ striatum, Gines et al. observed an increase in Akt activation and signaling via the phosphorylation of GSK3β (which is inhibited), which led to decreased β-catenin degradation. In this context, they observed an increase in the levels of the β-catenin target gene cyclin D1 [126].

Godin et al. firstly transfected MDCK and HEK293 cells with htt-480-17Q (expression of the N-term fragments of htt containing the first 480 aa with 17Q) or htt-480-68Q constructs. They observed a significant level of β-catenin with the htt-480-68Q construct with the two cell lines. β-catenin accumulated (in its phosphorylated form), but this accumulation does not lead to the transcriptional activation of the gene target. Then, they showed that the level of β-catenin is higher in different in vivo models of HD, such as drosophila, Hdh^CAG140^ and Hdh^Q111/Q111^ knock-in HD mice, and post-mortem samples of HD patients. [125].

Two more recent studies conducted on induced pluripotent stem cell (iPSC) cultures confirmed these observations. Smith-Geater et al. showed, in HD neuronal cells derived from iPSCs, a significant dysregulation of several members of the pathway, including the downregulation of the components of the β-catenin destruction complex and the increased expression of Transcription Factor 3 (TF3) and frizzled transcripts, as well as the increased expression of Wnt transcriptional targets such as CCND1 [129]. The upregulation of the gene target of the Wnt/β-catenin pathway was observed also by Lim et al. This team showed that components of the pathway are upregulated in iPSC-derived brain microvascular endothelial cells from HD patients, including ligands (WNT-3, -4, -6, -7B, and -10A), effectors (TCF3 and TCF4), and downstream targets (AXIN2 and APDCC1) [128]. They demonstrated that Wnt inhibition prevented angiogenic deficits and BBB impairment in their HD model.

In conclusion, except for two publications [124,127], it seems clear that the β-catenin level is higher in HD and Wnt inhibitors (indomethacin ICG-001, XAV-939) proved to be efficient in the studied models.

## 6. Hippo Pathway and Wnt Pathway in Other Neurodegenerative Diseases

The Hippo signaling pathway has recently emerged as one of the important pathways playing a pivotal role in central nervous system diseases, including neurodegenerative disorders beyond HD, such as Alzheimer’s disease (AD), amyotrophic lateral sclerosis (ALS), some retinal degeneration disorders (e.g., retinal detachment), cerebral ischemia-reperfusion, Alexander disease, dentatorubral–pallidoluysian atrophy (DRPLA), and Leber congenital amaurosis (LCA) [132,133,134]. Tanaka et al. also observed TRIAD phenotypes in AD with similar phenotypes. Moreover, they observed the presence of YAP in the Aβ aggregates, depriving YAP from the nucleus and LATS1 kinase activation in AD human cortical neurons. As in HD, the evidence suggests that the neurodegeneration could be linked to the Hippo pathway-dependent necrosis named TRIAD and, in particular, to the deprivation of YAP from the nucleus. Another similarity with HD is the positive effect of S1P to rescue ER instability in AD-iPSC-derived neurons [134]. TRIAD was observed also in ALS, but in this case, the pro-survival form of YAP (YAPdeltaC) decreases during neurodegeneration. Nevertheless, full-length YAP and p73 were preserved until the late symptomatic stage, and although the expression of total p73 also decreased with the progression of the disease, the ratio of phosphorylated p73 to total p73 increased during the late symptomatic stage, suggesting an important role of YAP-p73 dysregulation in the neurodegeneration of ALS [135]. Interestingly, the authors observed a significantly decreased level of nuclear YAP in the motor cortex of ALS patients compared to healthy controls and reduced YAP target gene expression in the motor cortex [136]. There is evidence that neuronal death in ALS could be rescued in the absence of Hippo signaling activation, and the lack of MST1 seems to be neuroprotective, suggesting the importance of the Hippo pathway in ALS [137,138]. The Hippo pathway was also found to be involved in retinal detachment, characterized by vision loss caused by photoreceptor cell death. In the mouse model of retinal detachment, MST2 was identified as a regulator of photoreceptor cell death, as MST-/- mice showed decreased photoreceptor cell death. Moreover, MST-/- mice demonstrated a reduction in proapoptotic molecules, such as PUMA, FAS, and activated caspase-3, and displayed suppressed nuclear relocalization of phosphorylated YAP [139]. This evidence suggests MST2 and the Hippo pathway as therapeutic targets for retinal detachment and also their neuroprotective role.

Recent evidence also evoked the involvement of the Hippo pathway in Parkinson’s disease (PD). Ahn et al. demonstrated that in PD, there is a downregulation of Netrin1 (NTN1), which is a secreted laminin-related protein involved in nervous system development, axon guidance, and growth [140,141,142]. NTN1 binds and phosphorylates netrin receptor UNC5B, leading to the apoptotic activation of UNC5B, responsible for dopaminergic neuronal loss. Due to the Netrin1 reduction, the activation of MST1 is increased and the kinase interacts and phosphorylates UNC5B, leading to decreased YAP levels and increasing cell death. This study showed that inhibiting UNC5B or the MST1 phosphorylation of UNC5B decreases neuronal apoptosis [143].

On the other hand, the Wnt pathway has also emerged to have an important role in neurodegenerative diseases, as came out from the constant increase in the literature on this topic. Indeed, numerous studies have shown that deregulations in the Wnt/β-catenin pathway are involved in the pathogenesis of neurodegenerative diseases such as AD, PD, and ALS. The Wnt/β-catenin pathway is known to have a neuroprotective effect against Aβ peptide in AD [144,145,146], and deregulations in this pathway are related to AD [147,148,149,150,151]. It was shown that Aβ fibrils have a destabilizing effect on β-catenin and, acting on Wnt/β-catenin activation, can antagonize these deleterious effects and counteract neurodegeneration and behavioral impairments; the activation can be both direct, through the Wnt-3a ligand, and indirect, through the inhibition of GSK-3β by lithium chloride (LiCl) [152,153,154]. Moreover, GSK-3β is responsible for the hyperphosphorylation of tau in AD. Wnt/β-catenin signaling plays a role in inhibiting tau phosphorylation and neurofibrillary tangles through the inhibition of GSK-3β [155]. On the other hand, DKK1, a Wnt antagonist, inhibiting the Wnt/β-catenin cascade, leads to tau hyperphosphorylation and neuronal death, and Aβ toxicity raises the DKK1 expression level, leading to increasing GSK-3β activity [156,157,158,159,160].

Similar evidence also emerged in PD. Mutations in the LRKK2 gene lead to the increased ability of LRRK2 to recruit GSK-3β and, as in AD, it phosphorylates the tau protein, leading to neurodegeneration [161]. Indeed, LRKK2 interacts with Dishevelled proteins, members of the Wnt family, and LRKK2 mutations are linked to the neurodegeneration and reduced activity of the Wnt canonical pathway [162,163]. Similar to AD, a high level of DKK1 and a low level of β-catenin with increased GSK-3β activity were also observed in PD, again, relating the downregulation of Wnt/β-catenin to neurodegeneration [164,165]. Another point in common with AD is that the administration of LiCl leads to the rescue of neurodegeneration due to the high levels of DKK1 [166]. Moreover, L’Episcopo et al. showed the role of Wnt1 signaling in midbrain dopaminergic neuron degeneration and self-repair in mice PD-injured midbrain, proposing a protective role for astrocytes, releasing Wnt1 ligand into the extracellular matrix, binding consequently the Fzd1 receptors, and activating the Wnt canonical cascade in order to protect neurons from neurodegeneration and promote neurogenesis [167,168].

Another neurodegenerative disease where the Wnt/β-catenin signaling pathway is involved is ALS. Studies in mice models demonstrated that several genes of the Wnt pathway, including ligands, receptors, co-receptors, and effectors are deregulated. It was observed that the upregulation of the Wnt1 ligand, Wnt3a, LRP5, and FZD receptors target genes such as Cyclin D1, c-myc, Fosl1, and Pitx2 [169,170]. Likewise, Chen et al. have shown an upregulation of mRNA and the protein of Wnt2 and Wnt7a correlated with the increased phosphorylation of GSK-3β [171]. Moreover, high levels of Wnt1 and Wnt3a were found in the spinal cords of ALS mice; these two ligands activate the canonical Wnt pathway, leading to the upregulation and nuclear translocation of β-catenin [170,172]. It was even found in human ALS patients that Wnt partners of the canonical cascade (Wnt5a and Fz2) are upregulated in the spinal cord [173]. Several Wnt components involved in cell growth regulation and proliferation have been found to be upregulated in ALS, which are likely to be involved in the pathogenesis of the disease and that probably have roles in the neurodegeneration of ALS.

Taken all together, these data show a strong relationship between the Wnt pathway and neurodegenerative diseases, and to deepen and go into detail on this relationship, it is useful to read Libro et al., 2016 and Serafino et al., 2020 [12,174].

All these data show the importance of the Hippo and Wnt pathways in neurodegenerative diseases. However, numerous studies are needed to fully understand the mechanism and role of these two pathways in neurodegenerative diseases, but the results are encouraging as they can represent a very important target in the treatment of these disorders.

## 7. Favoring YAP/TAZ-TEAD Interaction

The activation of the YAP/TAZ–TEAD interaction can be achieved through two main mechanisms: (i) decrease YAP/TAZ phosphorylation and therefore increase their nuclear localization, (ii) favor the YAP/TAZ–TEAD interaction with TEAD ligands.

The study of the overexpression of the representative members of different GPCR subgroups on YAP/TAZ phosphorylation by Yu et al. led to the discovery of the first activators and repressors of YAP/TAZ activity [50]. LPA and S1P (Figure 4) that bind to the G12/13 subgroup inhibit LATS1/2-inducing YAP dephosphorylation, favor its nuclear localization, and increase the production of CTGF and Cyr61, two target genes of the YAP/TAZ–TEAD interaction. These two drugs suppress balloon cell death in mouse primary cortical neurons and suppress ER instability and cell death in vivo in mutant Htt transgenic mice (R6/2: Htt-exon1-160Q) [123].

A cell-based screening of about 18,600 molecules at Tokyo Medical and Dental University led to the discovery of TT-10 [175], an activator of cardiomyocyte proliferation that activates the Wnt/β-catenin pathway through the inhibition of GSK3-β phosphorylation at Tyr-216 residue and also activates NRF2-mediated antioxidant and antiapoptotic effects. TT-10 (Figure 4) protects mice heart after ischemia injury.

The indirect inhibition of LATS1/2 can be achieved by the inhibition of MST1/2. XMU-MP-1 was identified by the high-throughput screening of an in-house compound library [176]. This molecule inhibits MST1 and MST2 with IC_50_ of 164 and 34 nM, respectively, by competing with ATP (10 µM). XMU-MP-1 (Figure 4) does not change the total YAP level but induces YAP nuclear localization in HepG2 cells at the micromolar level. It also increases the expression of the mRNA of CTGF and Cyr61 by about two-fold at 1 µM and about four-fold at 3 µM. At doses of 1 to 3 mg/kg (intraperitoneal injection), XMU-MP-1 increases intestinal repair, liver repair, and regeneration in both acute and chronic liver injury mouse models. XMU-MP-1 was used in several mice models to rescue the senescence of astrocytes and cognitive function [177], to promote the formation of glial scars, and to partially rescue demyelination and inflammatory infiltration [178,179].

To date, three inhibitors of Lats1/2 have recently been described. To compare the biological potency of these inhibitors, we indicate after IC_50′s_ of drugs the concentration of ATP at which the experiments have been conducted (the normal ATP cellular concentration is typically between 1–10 mM). The discrepancy between the ATP concentration used in enzymatic assays and the cellular concentration of ATP explains the difference in power between the enzymatic inhibition and the cellular activity.

TRULI (Figure 4) inhibits both LATS 1 and LATS2 with an IC_50_ of 0.2 nM (with 10 µM ATP) [180].

VT02956 has been identified through a high throughput screen of about 17,000 compounds in a program dedicated to identifying molecules for the treatment of breast cancer. Indeed, in this class of endocrine-resistant cancers, the expression of ESR1 (encoding ERα) is regulated by the expression of VGLL3, which recruits the NCOR2/SMRT repressor to the super-enhancer of the ESR1 gene. The inhibition of YAP and TAZ phosphorylation increases the nuclear import of YAP and TAZ, which compete with VGLL3. Finally, VGLL3 inhibits ESR1 expression. VT02956 in vitro inhibited LATS kinase with IC_50_ of 0.76 nM (LATS1; 3000 µM ATP) and 0.52 nM (LATS2; 6000 µM ATP) [38].

A cell-based, high-throughput screening developed for the identification of small molecules that promote cell proliferation (particularly in 3D conditions) led to GA-017 (Figure 4), a LATS kinase inhibitor that presents in vitro IC_50_ of 4.1 nM (LATS1; 400 µM ATP) and 3.9 nM (LATS2; 400 µM ATP) [181]. This compound promotes the formation of SKOV3 (ovarian adenocarcinoma) and A431 (squamous carcinoma) spheroids and mouse intestinal organoids by inducing YAP nuclear localization that induces significant increases in Ankrd1, Cyr61, and CTGF expressions at 5–20 µM. Using a similar approach on HEK293A-TEAD-luc cells, Shalhout et al. reported the activation of the YAP–TEAD interaction of PY-60 (Figure 4), a compound that liberates Annexin A2 from the membrane and in turn inhibits YAP phosphorylation [182]. The authors reported a micromolar affinity for Annexin A2 as measured by biolayer inferometry and the activation of YAP–TEAD target genes (Ankrd1, CTGF, and in less intensity, Cyr61) at 1–10 µM. This study revealed Annexin A2 as a new druggable component of the Hippo pathway that added complexity to the Hippo pathway.

Pobbati et al. reported the sole article on small TEAD ligands acting as activators of YAP/TAZ–TEAD rather than inhibitors [183]. Q2 and B22 (Figure 4) are 8-hydroxyquinolines substituted at position 7. Q2 is supposed to occupy the TEAD central pocket as shown by SPR (Surface Plasmon Resonance) on wt-hTEAD2 compared to the A231I mutant and increase expression levels of CTGF, Cyr61, and ANKRD1. In vivo, quinolinol Q2 accelerates cutaneous wound healing in mice as soon as day 7.

Adihou et al. designed an eicosapeptide 4E (derived from the VGL4 233-252 amino acid sequence by D to E replacement, leading to crosslinking via the lactamization of residues E235 and K250) (Figure 4) linked to the Tat sequence through a PEG2 linker [184]. Briefly, 4E binds to mTEAD4 (PDB code: 6SBA) at interface 2 and was found to be a greater inhibitor of VGLL4 than YAP and therefore activates the YAP–TEAD interaction, increases mRNA target gene levels (Cyr61, CTGF, ANKRD1, and SEPINE1) in human cardiomyocytes, and accelerates the wound healing of RKO cells.

Finally, recently, TRULI was used in an HD model of human neurulation to demonstrate that LATS could be a potential pharmacological target in HD [185].

## 8. Targeting Wnt/β-Catenin Signaling Pathway

Several approaches have been studied to target the Wnt/β-catenin pathway in neurodegenerative diseases. Synthetic and biological molecules that are under preclinical studies or under investigation in clinical trials have been recently reviewed [12]. The addressed diseases in clinical trials are AD, PD, ALS, and SMA (spinal muscular atrophy), but no investigation of a drug-targeting Wnt/β-catenin pathway for the treatment of HD has been conducted to date.

There is only one example of a drug that modified the Wnt pathway that was used in an HD model [126]. The non-steroid anti-inflammatory drug, indomethacin, known to downregulate β-catenin in cells, was used in primary cultures of striatal neurons transfected with wild-type huntingtin and polyQ-huntingtin (htt-480-17Q or htt-480-68Q) and showed a significant decrease in poly Q-huntingtin-induced neuronal toxicity vs. wt-HTT. The neuroprotective activity of indomethacin is not due to its sole anti-inflammatory activity but more likely to its multitarget effect since other AIDSs such as acetylsalicylate and rofecoxib did not show beneficial effects on transgenic HD mice [186,187].

A new emerging druggable target is the carboxylesterase, Notum, which is responsible for the depalmitoleoylation of Wnt in the extracellular space [188]. Its inhibition induces the activation of the Wnt/β-catenin pathway and could find applications in the field of cancers, osteoporosis, and neurodegenerative disorders [189]. It is expected that inhibitors of Notum could probably induce the nuclear localization of YAP/TAZ.

## 9. Discussion/Conclusions

In this review, we summarized the evidence that the Hippo and Wnt pathways are involved in Huntington’s disease. Post-mortem analysis of human HD brains showed the deregulation of several proteins of the Hippo pathway and particularly a decrease in nuclear YAP that correlates with the downregulation of the up-stream Hippo pathway regulators [121]. Similar results were obtained by Yamanishi et al. [122]. By contrast, the role of the Wnt pathway in HD pathogenesis has been proposed, on the sole basis of in vitro and in vivo HD models, and the analysis of human post-mortem brains is needed to confirm this hypothesis [126]. Mutant Htt was demonstrated to interfere with the β-catenin turnover by binding to members of the β-catenin destruction complex and with YAP-inducing BCD [125,126]. In these cellular models (mouse primary cortical neurons expressing Htt-104Q or MDCK and HEK293 cells transfected with Htt-17Q or Htt-68Q), mHtt affects the nuclear localization of both β-catenin and YAP.

As seen above, the Hippo pathway and the Wnt pathway are both implicated in cell proliferation, survival, differentiation, and apoptosis. Literature analysis shows that these two pathways are studied individually, whereas the copious evidence of crosstalk between them should convince researchers to consider them together. The implications of each of these pathways have recently emerged in various neurodegenerative diseases, and new deregulations of these two pathways have been observed in these disorders. However, to date, no article has reported a study on the two pathways in the context of one neurodegenerative disease. It would be of great interest to have a precise idea of how these two pathways are connected in pathological tissues. Even if the design of drugs aimed at activating the YAP/TAZ–TEAD interaction (indirectly or directly) is still in its infancy, biologists now have efficient tool compounds to study the effects of therapeutic intervention in disease models. New drugs acting on the Wnt pathway are now available, and it would be interesting to have a precise vision of their effects on both β-catenin and YAP/TAZ pathways. Indeed, acting on the targets offered by both the Hippo and Wnt pathways has yielded positive results both in vitro and in vivo so far, and potential new targets and methods of action relevant to both pathways are emerging.

Thus, new therapeutic approaches, including combination therapy, regarding these two pathways should be evaluated in the future, given the first promising results observed with some of these molecules described above.

## Figures and Tables

**Figure 1 cells-11-03631-f001:**
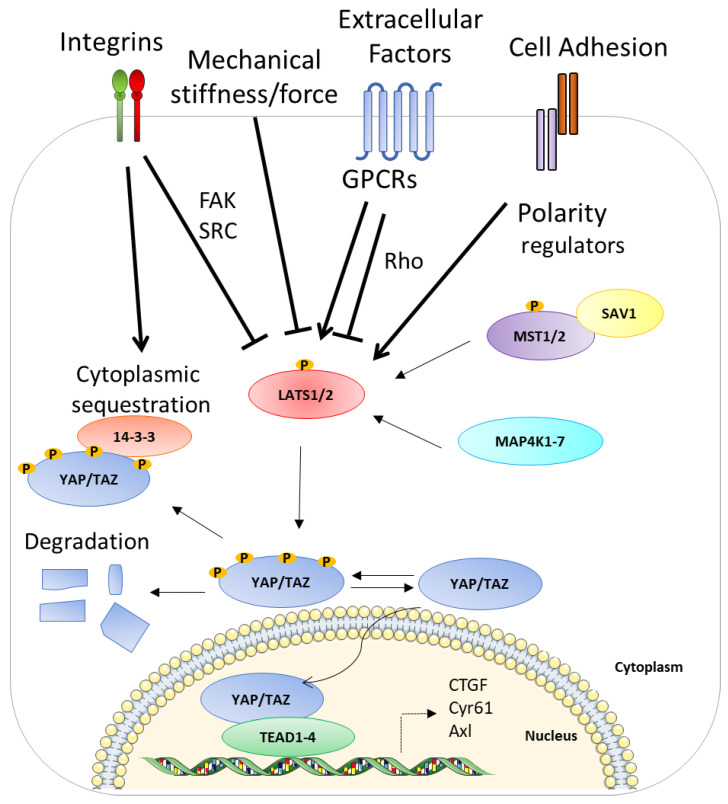
A schematic representation of the ON and OFF state of the Hippo pathway. The activation of the Hippo pathway leads to YAP phosphorylation and its subsequent sequestration and degradation. By contrast, the inactivation of the Hippo pathway allows YAP to enter the nucleus in order to interact with TEAD1–4 and activate their target genes. See text for details. The membrane image and DNA double helix are adapted from Servier Medical Art (smart.servier.com (accessed on 4 May 2017)) licensed under a Creative Commons Attribution 3.0 Unported License (SMART-Servier Medical ART. Available online: https://smart.servier.com/ (accessed on 4 May 2017)).

**Figure 2 cells-11-03631-f002:**
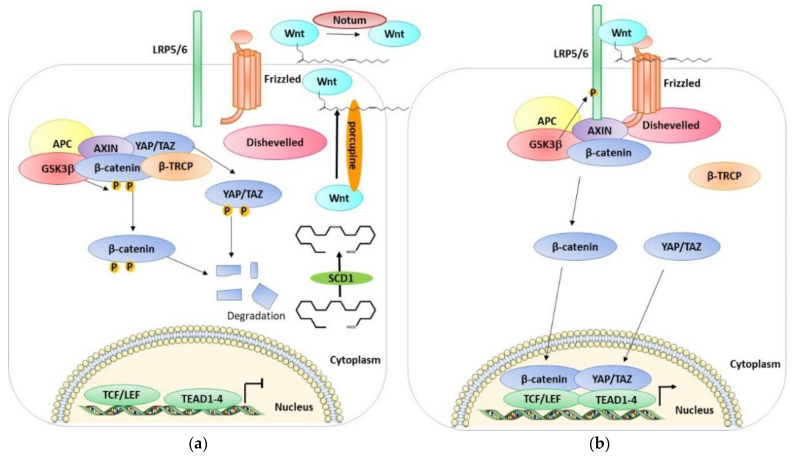
A schematic representation of the OFF and ON state of the Wnt pathway. (**a**) A scheme of the Wnt OFF state. The lack of Wnt ligands leads to the degradation of β-catenin, YAP/TAZ. Consequently, the level of β-catenin and YAP/TAZ in the cytoplasm is low, and they cannot enter the nucleus and activate their target genes. (**b**) A scheme of the Wnt ON state. Wnt ligands interacting with Frizzled inhibit the degradation of β-catenin and YAP/TAZ which accumulates in the cytoplasm and can enter the nucleus and interact with TCF/LEF and activate Wnt-responsive genes. See text for details.

**Figure 4 cells-11-03631-f004:**
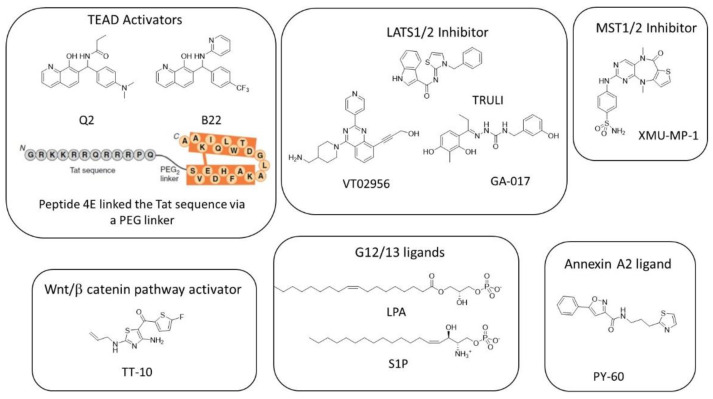
Structure of the known activators of YAP/TAZ–TEAD transcriptional activity.

## Data Availability

Not applicable.

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
