# Peer review of "Crosstalk between the Hippo Pathway and the Wnt Pathway in Huntington’s Disease and Other Neurodegenerative Disorders"

_cells, 2022, doi:10.3390/cells11223631_

Round 1

Reviewer 1 Report

This is an interesting and useful review article summarizing the current knowledge about the cross-talk between HIPPO and WNT pathways in neurodegenerative disease. I have rather minor comments to make:

A better editing of the English text is required. The images of Figures 2 and 3 are very difficult to discern.  The legend of Figure 2 is reversed (a is b and b is a).

Ref. PMID 27409837 concerning HIPP and oxidative stress should be included in the appropriate section of the text.  

Author Response

Responses to reviewer 1

This is an interesting and useful review article summarizing the current knowledge about the cross-talk between HIPPO and WNT pathways in neurodegenerative disease. I have rather minor comments to make:

A better editing of the English text is required.

This has been done by our colleague Dr Jean-Marc Taymans.

 The images of Figures 2 and 3 are very difficult to discern. 

The font size has been increased (from 10 to 14). We hope the figures will be more readable.

The legend of Figure 2 is reversed (a is b and b is a).

Corrections have been made.

Ref. PMID 27409837 concerning HIPP and oxidative stress should be included in the appropriate section of the text. 

The suggested article has been added in the article (lines 158-161)

Rajesh, K.; Krishnamoorthy, J.; Gupta, J.; Kazimierczak, U.; Papadakis, A.I.; Deng, Z.; Wang, S.;  Kuninaka, S.;  Koromilas, A.E. The eIF2α serine 51 phosphorylation-ATF4 arm promotes HIPPO signaling and cell death under oxidative stress Oncotarget , 2016, 7, 51044-51058. doi: 10.18632/oncotarget.10480.

Reviewer 2 Report

1. Suggestions on all figures: the fonts (gene manes) are relatively small.

2. Lines 464-466: Historically, the first reported YAP/TAZ phosphorylation repression was reported 464 by Yu et al. who studied overexpression of representative members of different GPCR 465 subgroups on YAP/TAZ activity [50]. This is confusing. The first YAP phosphorylation/repression was reported by Pan, Guan groups in 2007.

Author Response

Response to reviewer 2 

  1. Suggestions on all figures: the fonts (gene manes) are relatively small.

The font size has been increased (from 10 to 14)

  1. Lines 464-466: Historically, the first reported YAP/TAZ phosphorylation repression was reported 464 by Yu et al. who studied overexpression of representative members of different GPCR 465 subgroups on YAP/TAZ activity [50]. This is confusing. The first YAP phosphorylation/repression was reported by Pan, Guan groups in 2007.

The sentence has been modified as followed: “Study of the overexpression of representative members of different GPCR subgroups on YAP/TAZ phosphorylation by Yu et al. led to the discovery of the first activators and repressors of YAP/TAZ activity [50].” to avoid confusion.